# Learning a restricted Boltzmann machine using biased Monte Carlo sampling

**Nicolas Béreux[1★], Aurélien Decelle[1,2], Cyril Furtlehner[2] and Beatriz Seoane[1]**

**1** Departamento de Física Teórica, Universidad Complutense
de Madrid, 28040 Madrid, Spain.
**2** Université Paris-Saclay, CNRS, INRIA Tau team, LISN, 91190, Gif-sur-Yvette, France.

★ nicolas.bereux@gmail.com

## Abstract

Restricted Boltzmann Machines are simple and powerful generative models that can encode any complex dataset. Despite all their advantages, in practice the trainings are often unstable and it is difficult to assess their quality because the dynamics are affected by extremely slow time dependencies. This situation becomes critical when dealing with low-dimensional clustered datasets, where the time required to sample ergodically the trained models becomes computationally prohibitive. In this work, we show that this divergence of Monte Carlo mixing times is related to a phenomenon of phase coexistence, similar to that which occurs in physics near a first-order phase transition. We show that sampling the equilibrium distribution using the Markov chain Monte Carlo method can be dramatically accelerated when using biased sampling techniques, in particular the Tethered Monte Carlo (TMC) method. This sampling technique efficiently solves the problem of evaluating the quality of a given trained model and generating new samples in a reasonable amount of time. Moreover, we show that this sampling technique can also be used to improve the computation of the log-likelihood gradient during training, leading to dramatic improvements in training RBMs with artificial clustered datasets. On real low-dimensional datasets, this new training method fits RBM models with significantly faster relaxation dynamics than those obtained with standard PCD recipes. We also show that TMC sampling can be used to recover the free-energy profile of the RBM. This proves to be extremely useful to compute the probability distribution of a given model and to improve the generation of new decorrelated samples in slow PCD-trained models. The main limitations of this method are, first, the restriction to effective low-dimensional datasets and, second, the fact that the Tethered MC method breaks the possibility of performing parallel alternative Monte Carlo updates, which limits the size of the systems we can consider in practice.

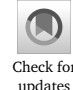

# 1   Introduction

In Physics, Restricted Boltzmann Machines (RBMs) are *only* a spin system or an Ising model in which the coupling matrix between spins has a bipartite topology. In Machine Learning, it is a famous generative model introduced many decades ago by Smolensky [1] as one of the first generative models capable of successfully learning complex/real datasets. It was later popularized by Hinton [2,3], who (at the time) proposed a viable algorithm for training these machines. After a brief period of notoriety in which RBMs were used as a pretraining method for larger neural networks [4,5], they fell into disuse after 2010 with the introduction of new generative models such as Generative Adversarial Network (GAN) [6] and Variational Autoencoder (VAE) [7]. Indeed, these models have a clear advantage: despite eventual instabilities of the training process, they do not rely on complex and costly Monte Carlo simulations. Nevertheless, RBMs have made a strong comeback in the field of Computational Biology in recent years, especially for purposes of pattern extraction [8–14].

RBMs remain an interesting generative model for many reasons. On the ML side, RBMs are simpler models than GAN or VAE. They have only one hidden layer, resulting in simple and potentially interpretable learned features. They can also learn complex datasets and are practical even when the number of samples of that dataset is limited. During the last decade, they have also regained the interest of physicists [8, 15–18], given their closeness to the well-known Ising model, for which computing the phase diagram in its many variations has been a long-standing hobby in the Statistical Physics community. In the case of RBMs, it is more difficult to obtain a relevant phase diagram, since the coupling constants of the trained model must be correlated due to the learning process. Nevertheless, the phase diagram obtained in simpler cases can at least be used as a guide to understand the macroscopic behavior of RBMs in the high temperature (linear) regime of the model. Lately, some recent work also has

attempted to analyze the learning dynamics of the models [19].

Despite all these efforts, training RBMs remains a difficult task, especially because of the instability of the training procedure, which is mainly related to the use of nonconvergent Markov Chain Monte Carlo (MCMC) processes to estimate the gradient during training, as stated in a recent work [20]. In this work, we present a new learning method that uses biased sampling Monte Carlo methods [21]. We will show that the classical alternative sampling MCMC method combined with Glauber Heat-Bath dynamics used in training RBMs is impractical for some datasets: the mixing time of the chain is simply too long. By using biased MC sampling, we can avoid this situation, especially at the beginning of learning. The drawbacks of this method is first that it breaks the conditional independence of the two layers of the RBM. Because of that, parallelization is less efficient and therefore restrain the size of the dataset one can be considered. Second, it needs to discretize an order parameter into possibly different directions, reducing its use when only few directions are needed.

This paper is organized as follows. We start with a definition of RBMs and their classical training procedures in Section 2. We then motivate our work by discussing the break of ergodicity phenomena encountered in RBMs trained with data following multimodal distributions. Then, in Section 3, we introduce the Tethered MCMC (TMC) sampling method and in Section 4, and its combination with the stochastic gradient ascent during the learning process in Section 4.3. In Section 6, we compare the new TMC training strategy with the standard training scheme in several artificial and real datasets.

## 2 Definitions

An RBM is an Ising spin model with pairwise interactions defined on a bipartite graph of two noninteracting layers of variables: the visible nodes, $\boldsymbol{v} = \{v_i, i = 1, ..., N_v\}$, represent the data, while the hidden nodes, $\boldsymbol{h} = \{h_a, a = 1, ..., N_h\}$, are the latent representations of this data and attempt to establish arbitrary dependencies between the visible units. Typically, the nodes are binary-valued in $\{0, 1\}$, yet Gaussian or arbitrary distributions on real-valued bounded support are also used, ultimately making RBMs adaptable to more heterogeneous datasets. Here we are concerned only with binary $\{0, 1\}$ variables for both the visible and hidden nodes. Other approaches that use truncated Gaussian hidden units [22] and provide an activation function of type ReLu for the hidden layer work well, but our experiments should not be affected by this choice. The energy function of an RBM is taken as

$$E[\boldsymbol{v}, \boldsymbol{h}; \boldsymbol{w}, \boldsymbol{b}, \boldsymbol{c}] = -\sum_{ia} v_i w_{ia} h_a - \sum_i b_i v_i - \sum_a c_a h_a, \tag{1}$$

with $\boldsymbol{w}$ being the weight matrix, and $\boldsymbol{b}$, $\boldsymbol{c}$ the visible, and hidden biases (or the magnetic fields in the Physics language), respectively. The Boltzmann distribution of such a model is then given by

$$p[\boldsymbol{v}, \boldsymbol{h} | \boldsymbol{w}, \boldsymbol{b}, \boldsymbol{c}] = \frac{\exp(-E[\boldsymbol{v}, \boldsymbol{h}; \boldsymbol{w}, \boldsymbol{b}, \boldsymbol{c}])}{Z}, \tag{2}$$

where $Z$ is the partition function. RBMs are usually trained using gradient ascent of the log-likelihood function, $\mathcal{L}$, of the training dataset, $\mathcal{D} = \{\boldsymbol{v}^{(1)}, \cdots, \boldsymbol{v}^{(M)}\}$, being $\mathcal{L}$ defined as

$$\mathcal{L}(\boldsymbol{w}, \boldsymbol{b}, \boldsymbol{c} | \mathcal{D}) = M^{-1} \sum_{m=1}^{M} \ln p(\boldsymbol{v} = \boldsymbol{v}^{(m)} | \boldsymbol{w}, \boldsymbol{b}, \boldsymbol{c}) = M^{-1} \sum_{m=1}^{M} \ln \sum_{\{\boldsymbol{h}\}} e^{-E[\boldsymbol{v}^{(m)}, \boldsymbol{h}; \boldsymbol{w}, \boldsymbol{b}, \boldsymbol{c}]} - \ln Z.$$

The gradient of $\mathcal{L}$ is then composed of two terms: the first one accounts for the interactions between the RBM's response and the training set, and the same for the second, but using the

samples drawn by the machine itself. The expression of the $\mathcal{L}$ gradient w.r.t. all the parameters is given by

$$\frac{\partial \mathcal{L}}{\partial w_{ia}} = \langle v_i h_a \rangle_\mathcal{D} - \langle v_i h_a \rangle_\mathcal{H}, \tag{3}$$

$$\frac{\partial \mathcal{L}}{\partial b_i} = \langle v_i \rangle_\mathcal{D} - \langle v_i \rangle_\mathcal{H}, \tag{4}$$

$$\frac{\partial \mathcal{L}}{\partial c_a} = \langle h_a \rangle_\mathcal{D} - \langle h_a \rangle_\mathcal{H}, \tag{5}$$

where

$$\langle f(\boldsymbol{v}, \boldsymbol{h}) \rangle_\mathcal{D} = M^{-1} \sum_m \sum_{\{\boldsymbol{h}\}} f(\boldsymbol{v}^{(m)}, \boldsymbol{h}) p(\boldsymbol{h} | \boldsymbol{v}^{(m)}),$$

denotes an average of an arbitrary function $f(\boldsymbol{v}, \boldsymbol{h})$ over the dataset, and $\langle f(\boldsymbol{v}, \boldsymbol{h}) \rangle_\mathcal{H}$, the average over the Boltzmann measure in Eq. (2).

A large part of the literature on RBMs focuses on schemes for approximating the r.h.s. of Eqs. 3-5, usually referred to as *negative term* of the gradient. Theoretically, these terms can be computed to arbitrary accuracy using parallel MCMC simulations, as long as the number of MC steps is large enough to ensure a proper sampling of the equilibrium configuration space. The most popular approximation schemes for computing the negative part are: (i) the Contrastive Divergence (CD) introduced decades ago by Hinton [3], or its later refinement the (ii) Persistent CD (PCD) [23]. In both cases, the negative part of the gradient is estimated after a few $k$ sampling steps, typically $k \sim \mathcal{O}(1)$, and the differences are related only to different clever ways of initializing the Markov chains. In the CD recipe, the parallel MCMC simulations are initialized from the same samples of the mini-batch that are used to compute the positive part of the gradient, while in the PCD recipe, the final configurations of each Markov chain are stored from one parameter update to the next to initialize the subsequent chain. In practice, the consequences of the lack of convergence of all these MCMC processes under these approximations during learning have very rarely been studied in the Literature, and it has recently been shown to have dramatic and nonmonotonic effects on the generation performance of RBMs [20]. In fact, this work has shown that the CD method is generally a very poor training method because it fits models whose equilibrium distribution is drastically different from the dataset distribution. In contrast, the PCD recipe appeared to fit RBMs with good equilibrium properties in image datasets, but the quality of the samples generated by these machines was limited by the number of sampling steps $k$ used during learning.

A closer examination of the PCD scheme shows us examples where this strategy also fails completely in learning good model parameters. For example, it was shown in [24] that simple artificial datasets with low effective dimension, such as well-separated point clusters, cannot be learned correctly with PCD. One might naively think that such a simple dataset should be straightforward to learn with an RBM, considering that this problem is easily solved with a Gaussian Mixture model. On the contrary, the main problem of such "clustered" datasets is that it is very difficult to ergodically sample the configuration space of multimodal distributions. In a one-dimensional case, i.e., clusters separated along a spatial direction, the typical probability density profile along this direction should correspond to well-fitted Gaussian peaks centered at each cluster center and separated by regions with essentially zero probability. It is well known that in such a case the MCMC chains can take an extremely long time to jump from one cluster to another. In Physics, we would speak of large free energy barriers to surmount, and it is exactly the same phenomenon encountered in the vicinity of a first order phase transition. This shows that a learning procedure that relies on parallel MCMC samplings to estimate the correlations of the model should be extremely costly or even fail miserably when trying to learn *clusterized* data.

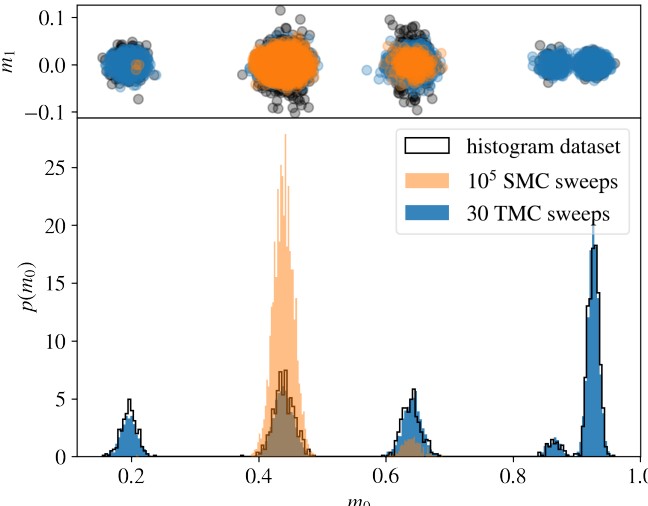

Figure 1: We illustrate the difficulties of sampling a multimodal distribution with standard Glauber MC moves of a dataset of 5 aligned point clusters in high dimensions. On the top, we show the projection of the dataset/generated samples along the first ($m_0$) and the second ($m_1$) dataset's principal components. Each dot is one sample. In the bottom, we show the histogram of all these $m_0$. The original dataset is displayed in black, and the generated datasets are shown in blue or orange, depending on the algorithm used to sample the configuration space of the exactly trained RBM from Ref. [24], from random initializations. Orange samples are obtained after $10^5$ standard Glauber MC (SMC) steps and blue samples, after 30 TMC sweeps.

Many interesting datasets form compact clusters after dimensional reduction, which is the typical case of genome or protein sequence data, for example. The previous discussion anticipates that it must not only be difficult to obtain reliable and effective models for such datasets using standard learning schemes (such as CD /PCD), but also to evaluate the quality of a given trained model. Indeed, standard sampling techniques are unable to equilibrate, which means that a convergent generation of samples may be prohibitively long. As a consequence, approximate techniques typically used to compute $\mathcal{L}$, such as Importance Sampling [25], must be completely inaccurate on such datasets.

The numerical problems associated with the apparition of multiple coexisting phases have long been known in Physics or Chemistry. One of the possible solutions is to break the metastability with external constraints. This idea can be easily implemented using biased MC or reweighting methods, such as the Umbrella Sampling method [26], microcanonical methods [27, 28], or the Tethered MC (TMC) method [29–31], which is adapted here to training and generation sampling of RBMs.

# 3 Breakdown of ergodicity

Multimodal distributions are notoriously hard to sample because simulations remain trapped in one or several metastable free energy minima for a very long time and fail to visit the entire configuration space. We illustrate this difficulty in Fig. 1. For this purpose, we considered an artificial dataset with 5 point clusters in 1000 dimensions, where all 5 are aligned in a single dimension. For this dataset, it is possible to train an RBM exactly as described in Ref. [24]. We use this perfectly trained RBM to generate new data using either standard MCMC moves (using

Glauber-Heath-Bath dynamics), – a sampling procedure that will henceforth be referred to as SMC –, or the TMC method that we will introduce later. In both cases, the term MC sweeps refers to an attempt to update all $N_{\mathrm{v}}$ and $N_h$ variables of the model.

Before proceeding with the discussion of Fig. 1, let us briefly discuss the projections we use to compare the original and generated datasets. We begin by extracting the principal components of the dataset $\boldsymbol{\omega}_\alpha$ with $\alpha = 0, \cdots N_{\mathrm{v}}-1$ according to standard Principal Component Analysis (PCA). Thus, $\boldsymbol{\omega}_0$ marks the direction of maximum variance of the dataset, $\boldsymbol{\omega}_1$ the next, and so on (with the constraint that all $\{\boldsymbol{\omega}_\alpha\}$ are orthonomal). Now we use these directions to project each data point, $\boldsymbol{v}^{(m)}$,

$$m_\alpha^{(m)} = \frac{\boldsymbol{v}^{(m)} \cdot \boldsymbol{\omega}_\alpha}{\sqrt{N_{\mathrm{v}}}}, \tag{6}$$

which defines the datum's *magnetization* along the $\alpha$ principal direction. In Fig. 1, we show the first two magnetizations, $m_0^{(m)}$ and $m_1^{(m)}$, and the distribution $p(m_0^{(m)})$ for the different sets of data. Note that one needs just one direction to split up the 5 clusters (they are aligned by construction), and it is straightforward to capture such a direction with the PCA. The choice of using the PCA in general will be justified in Sect. 5. Clearly, from a random initialization, local SMC moves get trapped in just one of the 5 data clusters even after performing $10^5$ MC sweeps, thus failing to visit (and generate) data in the rest of the clusters. The breakdown of ergodicity is easy to understand here: in order to jump from one cluster to another with local moves, the trajectory must visit configurations with an intermediary $m_0$, and such configurations have extremely low probability. In the TMC sampling, one can enforce the simulations to visit these rare but crucial intermediate states by introducing external constraints. As a result, this sampling method is able to generate samples in all the five clusters with the correct associated weights.

## 4 Sampling with the tethered MC method

We use the TMC sampling method [29, 31] to sample efficiently the configuration space of clustered models. TMC is a refinement over the popular Umbrella Sampling method [26] that permits simplifying notably the reconstruction of canonical expected values. This simplification will be crucial to combine this sampling technique with the stochastic training dynamics. In the TMC method, a soft constraint $m(\boldsymbol{v}) \sim \hat{m}$[1] is introduced in the partition function $Z$ via the identity

$$\sqrt{\alpha/2\pi} \int_{-\infty}^{\infty} d\hat{m} \, e^{-\frac{\alpha}{2}(m(\boldsymbol{v})-\hat{m})^2} = 1,$$

where $\alpha$ controls the strength of the constraint. In this way, $Z$ can be written in terms of the TMC *effective potential*, $\Omega(\hat{m})$,

$$Z = \int_{-\infty}^{\infty} d\hat{m} \, e^{-\alpha\Omega(\hat{m})}, \tag{7}$$

with

$$e^{-\alpha\Omega(\hat{m})} = \sqrt{\frac{\alpha}{2\pi}} \sum_{\{\boldsymbol{v},\boldsymbol{h}\}}{}' e^{-E(\boldsymbol{h},\boldsymbol{v})-\frac{\alpha}{2}[m(\boldsymbol{v})-\hat{m}]^2}. \tag{8}$$

Then, the probability of $\hat{m}$ is given by

$$p(\hat{m}) = \frac{e^{-\alpha\Omega(\hat{m})}}{Z}. \tag{9}$$

---

[1]Technically the global constraint could perfectly concern both the visible and the hidden units, $m(\boldsymbol{h},\boldsymbol{v}) \sim \hat{m}$, or only the hidden $m(\boldsymbol{h}) \sim \hat{m}$ too.

Clearly, when $\alpha \to \infty$, $p(\hat{m})$ recovers the canonical probability of finding $m(\boldsymbol{v}) = \hat{m}$, that is, $\langle \delta(m(\boldsymbol{v}) - \hat{m}) \rangle$. Finite values of $\alpha$ soften this constraint, which is necessary to facilitate sampling. We can use this probability to define a new ensemble, the *tethered ensemble*, where $m(\boldsymbol{v}) \sim \hat{m}$. Then, the expected value of an observable $O(\boldsymbol{h}, \boldsymbol{v})$ in this ensemble is

$$\langle O \rangle_{\hat{m}} = \frac{\sum_{\{\boldsymbol{h},\boldsymbol{v}\}} O(\boldsymbol{h},\boldsymbol{v}) \, \omega(\boldsymbol{h},\boldsymbol{v},\hat{m})}{\sum_{\{\boldsymbol{h},\boldsymbol{v}\}} \omega(\boldsymbol{h},\boldsymbol{v},\hat{m})}, \tag{10}$$

with

$$\omega(\boldsymbol{h},\boldsymbol{v},\hat{m}) = e^{-E(\boldsymbol{h},\boldsymbol{v}) - \frac{\alpha}{2}[m(\boldsymbol{v}) - \hat{m}]^2}. \tag{11}$$

Now, the principal innovation of the TMC method is realizing that

$$\Omega'(\hat{m}) = \frac{d\Omega}{d\hat{m}} = \langle \hat{m} - m(\boldsymbol{v}) \rangle_{\hat{m}}, \tag{12}$$

which means that $\Omega(\hat{m})$ can be recovered up to any desired precision, from a numerical integration of Monte Carlo averages of several independent simulations at fixed $\hat{m}$

$$\Omega(\hat{m}) - \Omega(\hat{m}_{\min}) = \int_{\hat{m}_{\min}}^{\hat{m}} \frac{d\Omega}{d\hat{m}'} d\hat{m}' = \int_{\hat{m}_{\min}}^{\hat{m}} \langle \hat{m}' - m(\boldsymbol{v}) \rangle_{\hat{m}'} d\hat{m}'. \tag{13}$$

So does $p(\hat{m})$,

$$\frac{e^{-\alpha\Omega(\hat{m})}}{\int_{\hat{m}_{\min}}^{\hat{m}_{\max}} e^{-\alpha\Omega(\hat{m}')} d\hat{m}'} \to p(\hat{m}), \tag{14}$$

as long as, $\hat{m}_{\min}$ and $\hat{m}_{\max}$, the two extrema of the integration interval, are safely chosen so that $e^{-\alpha\Omega(\hat{m})}$ is essentially zero beyond that range. Also, $p(\hat{m}) \approx \langle \delta(m(\boldsymbol{v}) - \hat{m}) \rangle_{\mathcal{H}}$ for large $\alpha$. In this work, we use typically $\alpha \sim 10^4$, which means that the approximation is rather good. In addition, we can exploit the TMC strategy to estimate the free energy barrier between two metastable states.

Interestingly, it is straightforward to show that the canonical average of $O$, $\langle O \rangle_{\mathcal{H}}$, that is, the average with respect to the Boltzmann measure of Eq. (2) (the quantity one wants to compute in practice), can be recovered using the following identity:

$$\langle O \rangle_{\mathcal{H}} = \int_{-\infty}^{\infty} \langle O \rangle_{\hat{m}} \, p(\hat{m}). \tag{15}$$

In other words, model averages can be recovered up to any desired precision just by integrating TMC averages (with no approximation involved). Such a turnover between ensembles is only useful if sampling ergodically the TMC measure (11) is much easier than in the original unconstrained problem (2). This is particularly true when only one phase is stable at a given $\hat{m}$. In the absence of metastabilities, thermalization is extremely fast. This means that one can use the TMC weight of Eq. (11) to avoid phase coexistence (and long time dependencies) in your MC simulation.

## 4.1 Example: TMC Sampling of the 5 aligned cluster dataset

Let us come back to the 5 clusters dataset example used to illustrate the ergodicity problem (discussed around Fig. 1). In the light of the preceding discussion, it is now clear that we can efficiently sample the whole configuration space running TMC simulations with $\hat{m}$ running in the range of observed $m_0$. We have summarized the TMC sampling procedure for this example in Fig. 2. The first step is to discretize the range of $m_0$ needed to integrate numerically $\Omega'(\hat{m})$

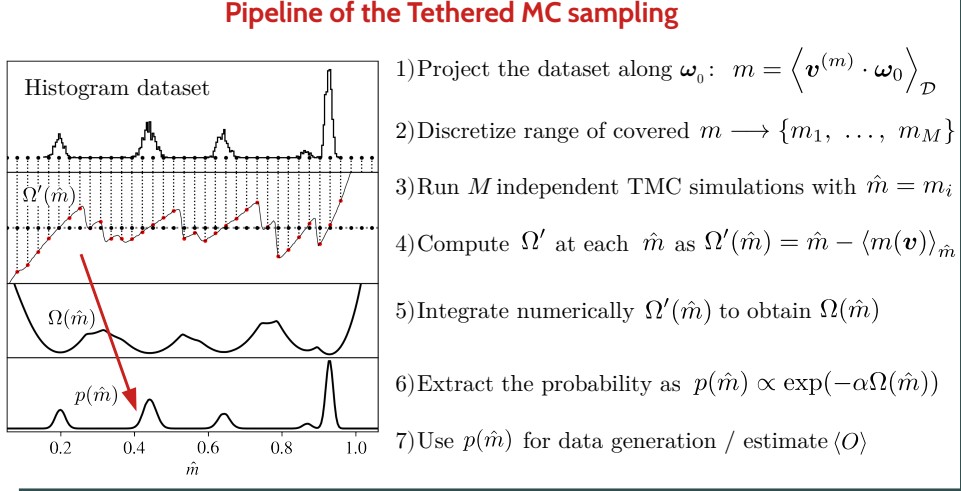

**Pipeline of the Tethered MC sampling**

1) Project the dataset along $\boldsymbol{\omega}_0$: $m = \left\langle \boldsymbol{v}^{(m)} \cdot \boldsymbol{\omega}_0 \right\rangle_{\mathcal{D}}$

2) Discretize range of covered $m \longrightarrow \{m_1, \ldots, m_M\}$

3) Run $M$ independent TMC simulations with $\hat{m} = m_i$

4) Compute $\Omega'$ at each $\hat{m}$ as $\Omega'(\hat{m}) = \hat{m} - \langle m(\boldsymbol{v}) \rangle_{\hat{m}}$

5) Integrate numerically $\Omega'(\hat{m})$ to obtain $\Omega(\hat{m})$

6) Extract the probability as $p(\hat{m}) \propto \exp(-\alpha\Omega(\hat{m}))$

7) Use $p(\hat{m})$ for data generation / estimate $\langle O \rangle$

Figure 2: We describe the basic steps of the TMC sampling procedure, which can be used either to generate equilibrium samples of the model or to estimate the log-likelihood gradient during training.

in (13) and to obtain $p(\hat{m})$ (14). To do this, we need to select a good interval of $\hat{m}$, that is, a $\hat{m}_{\min}$ and $\hat{m}_{\max}$. We can do this using the minimum and maximum projection of the dataset along $\boldsymbol{\omega}_0$, plus a safety border to ensure zero probability beyond these points. In order to avoid scaling problems between RBMs models when defining $\hat{m}$, we have always normalized the data projections conveniently, so that they lie in the interval $[0, 1]$, and considered extra borders of $\pm 0.1$ or $\pm 0.2$ to define the range of $\hat{m}$. Then, we discretize uniformly the interval $[\hat{m}_{\min}, \hat{m}_{\max}]$ using $N_{\hat{m}}$ points (for the sampling in Fig. 2 we considered 250 discretization points, even though we show much less in the sketch to enhance the comprehension). Then, we run independent TMC simulations at each of these $\hat{m}$ values. In general, we consider several parallel runs for each $\hat{m}$ to estimate $\Omega'(\hat{m})$ in (12). This is much faster than iterating TMC many times because they can run in parallel. Note that, the TMC weight of Eq. (11) breaks the independence of the visible units provided by the bipartite lattice structure, which means that we lose the possibility of updating all visible variables at the same time (the so-called alternating Gibbs sampling) which makes the TMC simulations much slower than the unconstrained model. Once we have computed $\Omega'(\hat{m})$ for all the discretized values of $\hat{m}$, we integrate this function numerically to extract $\Omega$ and $p(\hat{m})$. We compare the $p(\hat{m})$ extracted following this pipeline[2] with the histogram of $m_0$ obtained from the dataset in Fig. 3, showing a perfect match between both distributions. Finally, one can use this $p(\hat{m})$ to improve the estimation of the $\mathcal{L}$ gradients via Eq. (15) during the learning process, as we will discuss below, or just to generate samples from our model. In this second case, the procedure is simple, draw a set of $\hat{m}_{\text{sampling}}$ values from the $p(\hat{m})$ distribution, and run independent and short TMC runs (initialized at random or other non-informative configuration) at these $\hat{m}$s. The distribution of the projections of the samples generated using this procedure were shown in blue in Fig. 1, showing again a perfect match with the distribution of the original dataset.

## 4.2 Generalization to multiple order parameters

The method can easily be generalized to $k$ constraints. Let $\boldsymbol{m} = (m_1, \cdots, m_k)$ represent these parameters. The method remains the same: a constraint is imposed on the parameters at the

---

[2]Actually $p(\hat{m})$ versus $\langle m_0(\boldsymbol{v}) \rangle_{\hat{m}}$ for the whole range of $\hat{m}$ simulated.

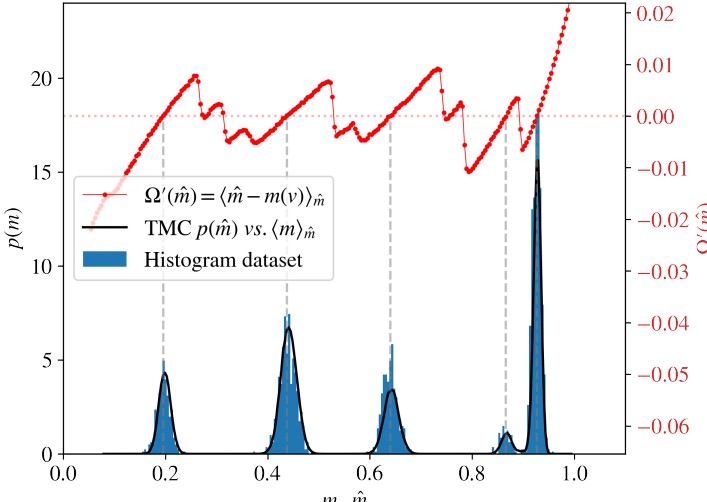

Figure 3: Reconstruction of the probability distribution profile of the dataset using TMC sampling. In red we show the derivative of the TMC effective potential, $\Omega'(\hat{m})$, obtained from TMC simulations at fixed values of $\hat{m}$ and $\alpha = 2 \cdot 10^4$. In black, we show the Tethered probability $p(\hat{m})$ obtained from the numerical integration of $\Omega(\hat{m})$ plotted against $\langle m \rangle_{\hat{m}}$. We compare this probability with the histogram of the projection of the dataset along $\boldsymbol{\omega}_0$. An almost perfect match is seen, even though the two probabilities should only match in the limit $\alpha \to \infty$.

same time, hence the weights of the TMC ensemble are now given by :

$$\omega(\boldsymbol{h}, \boldsymbol{v}, \hat{\boldsymbol{m}}) = e^{-E(\boldsymbol{h},\boldsymbol{v})-\frac{\alpha}{2}\sum_{l=1}^{k}(\hat{m}_l - m_l)^2} \, . \tag{16}$$

The gradient of the potential is now

$$\boldsymbol{\nabla}\Omega(\hat{\boldsymbol{m}}) = \langle \hat{\boldsymbol{m}} - \boldsymbol{m} \rangle_{\hat{\boldsymbol{m}}} \, , \tag{17}$$

and the potential can be recovered at fixed $\hat{\boldsymbol{m}}$ through the numerical integration of $\boldsymbol{\nabla}\Omega(\hat{\boldsymbol{m}})$:

$$\Omega(\hat{\boldsymbol{m}}) = \Omega(\hat{\boldsymbol{m}}_{\mathbf{min}}) + \int_{\mathcal{C}} \boldsymbol{\nabla}\Omega(\hat{\boldsymbol{m}}')d\hat{\boldsymbol{m}}' \, , \tag{18}$$

where $\mathcal{C}$ represents any path leading from $\hat{\boldsymbol{m}}_{\mathbf{min}} = (\hat{m}_{1_{\mathbf{min}}}, \cdots, \hat{m}_{k_{\mathbf{min}}})$ to $\hat{\boldsymbol{m}}$. Finally, $p(\hat{\boldsymbol{m}})$ can be recovered analogously to the single order parameter case (14).

## 4.3 Training RBM with the Tethered Monte Carlo method

The TMC method is therefore very efficient to sample quickly isolated clusters. It is then straightforward to use it to compute the negative term of the gradient eqs. (3-5). For this goal, we need to run several independent simulations at fixed $\hat{m}$ and estimate $\langle s_i \rangle_{\hat{m}}$, $\langle h_a \rangle_{\hat{m}}$ and $\langle s_i h_a \rangle_{\hat{m}}$. We then recover the unconstrained negative term of the gradient using a numerical integration as in (15). In order to improve statistics in the gradient computation, we average each observable $O$, not only using $M$ parallel samples (running each at fixed $\hat{m}$), but also in time $T$ thermal history. This means that each $\langle O \rangle_{\hat{m}}$ is estimated by

$$\overline{O}^{\hat{m}} = \frac{1}{M}\sum_{k=1}^{M}\frac{1}{T}\sum_{t=1}^{T}O^{(k)}(t), x \, , \tag{19}$$

during the simulation. The code we used for the article can be freely downloaded on GitHub.

# 5 Learning dynamics and principal components of the dataset

RBMs have been thoroughly studied during the last decade in the context of statistical physics (see Ref. [18] for a recent review on the topic) and very important progresses have been made related to the understanding of their thermodynamics and learning mechanisms [8,15,17,19]. In particular, we now know that the RBMs pattern encoding process is triggered by learning the PCA of the dataset, i.e. the directions $\{\boldsymbol{\omega}_\alpha\}$ introduced in Sect. 3. At the beginning of the learning, when the machine is in a paramagnetic state [16,32,33], the learning dynamics can be easily understood when decomposing the RBM's weight matrix onto its singular value decomposition (SVD), $\boldsymbol{w} = \boldsymbol{U}\boldsymbol{\Sigma}\boldsymbol{V}^T$, and projecting the gradient equations 3-5 onto each of these modes. In the regime where all the elements of $\boldsymbol{w}$ are small, it can be shown that the first columns of $\boldsymbol{V}$ progressively align with the principal components of the dataset one by one, a process that can be easily followed through the emergence of new eigenvalues in the SVD. At larger learning times, the linear approximation of the gradient is no longer valid and the relation between the dataset's principal directions and the ones of the weight matrix is no longer true.

This means that in the early times of the learning process, the RBM passes from being in a paramagnetic phase to a ferromagnetic phase, with non-zero magnetizations $m_\alpha$ (recall (6)) along a growing number of directions $\alpha$, related to the number of the dataset's principal components encoded in the SVD rotation matrix $V$ and the number of hidden nodes of the RBM. We also know that the relevant directions to project the data are for some time $\{\boldsymbol{\omega}_\alpha\}$, and therefore related to the eigenmodes of the weight matrix. These theoretical results suggest that $m_\alpha$ must be good order parameters to control thermalisation (and to break metastabilities) at least for some time. The main drawback here, is that the longer the learning, the higher the number of constraints needed to sample ergodically the phase space. In the following, we will use the TMC method to constraint the value of the magnetization along 1 or 2 fixed PCA directions.

# 6 Results

In this section, we will compare the quality and the dynamics of RBMs trained using either the standard Glauber MC updates (SMC), or our new TMC sampling strategy to estimate the negative part of the $\mathcal{L}$ gradient. In both cases, we will always keep the PCD initialisation scheme to enhance as much as possible the thermalisation during the training and perform the same number of Gibbs steps between each parameter update. We further designate SMC-RBM to machines trained with the usual procedure (SMC), and TMC-RMB, to machines trained with TMC. We begin with results on artificial datasets to end with applications in real ones.

## 6.1 Artificial datasets

To illustrate the TMC learning strategy, we will first analyze the learning process of artificial datasets. At a second stage, we will move to real clustered datasets. We will consider 2 different high-dimensional datasets having $N_{\rm v} = 1000$ visible variables both. The first one, consists of two clusters supported on a one-dimensional subspace given by the vector $\boldsymbol{u} = \boldsymbol{1}/\sqrt{N_{\rm v}}$. The second, is made of three separated clusters supported on a two-dimensional subspace. In both cases not only one (or two directions) is sufficient to separate the clusters, but they also lie in a low-dimensional subspace: their extension to the other dimensions is only due to the noise in the creation of the dataset. Both datasets were designed to exaggerate a clustered nature, this means that, points are generated so they form compact clusters separated by large empty

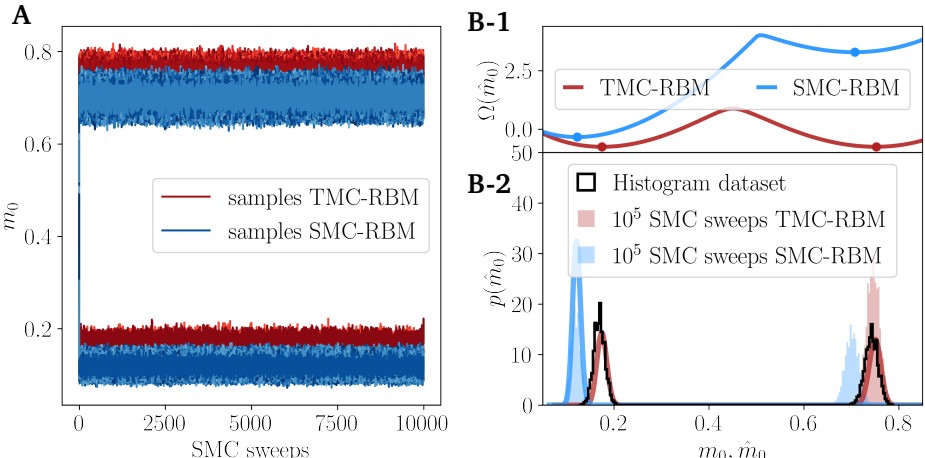

Figure 4: We compare RBMs trained with two different sampling procedures. We analyze models trained for 481 parameter updates and a learning rate of $10^{-2}$ for the TMC-RBM and $10^{-4}$ for the SMC-RBM. The SMC-RBM is trained with PCD and 10 MC sweeps of Glauber dynamics (the standard procedure, SMC), and the TMC-RBM is trained with PCD and 10 TMC1D sweeps fixing the projection $m_0$. Once both machines are trained, we can sample their configuration space with SMC moves in **A** or TMC moves in **B**. In **A** we show the evolution of $m_0(t)$ for 1000 independent simulations initialized at random (in blue we show the samplings of SMC-RBM, in red of the TMC-RBM). Fig. **A** shows that ergodicity is completely broken: Markov chains remain trapped in the state closest to their random initialization, and not a single jump from one state to the other is ever observed. This means that thermalization is not even close, and it is impossible to properly estimate the relative height of the two peaks from SMC simulations. In **B**, we instead take both machines under the microscope using the TMC algorithm to properly explore the equilibrium measure. The TMC results are shown in thick solid lines. In **B-1** we show the TMC effective potential $\Omega(\hat{m})$, which has 2 relative minima for both models. We mark the position of these minima with large dots. We can easily see that they coincide with the mean projection of the two states we sampled with SMC in **A**. However, in the case of the SMC-RBM, the left minima is much lower, which translates into a much higher TMC probability once $p(\hat{m})$ is computed in **B-2**. It is clear that none of the reconstructed equilibrium probabilities match the histograms obtained with the non-ergodic SMC sampling.

regions. In both cases, points can be easily split up in clusters by projecting the dataset on the first ($\boldsymbol{\omega}_0$, for the 1D case) or the first two ($\boldsymbol{\omega}_0$ and $\boldsymbol{\omega}_1$ for the 2D case) principal components of the dataset.

For this type of datasets, we expect a simple spin-flip type MCMC algorithm to fail dramatically when it comes to sampling ergodically a correctly trained RBM, because jumping from one cluster to another requires surpassing a very large free energy barrier (note that the region between clusters has essentially zero probability by construction). Such a problem is likely to make learning a good RBM on this type of dataset impossible in practice. Indeed, one would need a prohibitive number of MC sweeps to compute the negative term of the gradient, and the chains would never reach equilibrium even with standard methods such as PCD.

### 6.1.1 One-dimensional dataset

As anticipated, the 2 trained RBMs cannot be sampled ergodically using SMC moves as we illustrate in Fig. 4-A: simulations get trapped in one of the two metastable states from the very first MC sweeps and remain there at least until $10^5$ MC sweeps. With this dataset, the learning of the SMC-RBM is very unstable, therefore we used different learning rate for the two RBMs: $10^{-2}$ for the TMC-RBM and a much slower one, $10^{-4}$, for SMC-RBM. At this point, it is important to stress that one manages to generate points belonging to two different clusters only thanks to the random initialization of the Markov chains. In particular, we initialize each $v_i$ as $\{0, 1\}$ with 50% of probability, which means that initial configurations have in average $m_0 \sim 0.5$, which depending on the profile of the potential $\Omega$, tend trap the chains in the nearest free-energy minima, independently on its relative depth. Despite both RBMs generating data in two separate clusters, such clusters are not the same for the 2 machines. This is much clearer if we compare the distribution of the magnetizations $m_0$ of the generated samples after $10^5$ SMC sweeps with each RBM, with those of the dataset, see Fig. 4-B2. Clearly the SMC-RBM generates samples in the wrong region (with wrong values of $m_0$) and this situation does not improve at all if we continue the learning for much longer. On the contrary, the samples generated with the TMC-RBM are at the right position. On Fig. 4-B2, we see only one (red) peak since the SMC is incapable of jumping from one cluster to the other, still the second peak would match the position of the dataset as can be checked on the panel A of the same figure.

As justified above, we can efficiently sample the phase-space of these two trained machines using the TMC sampling method (using simulations at fixed $m_0$). We show the results in plain lines in Fig. 4-B. In Fig. Fig. 4-B1, we show the Tethered effective potential $\Omega$, as function of $\hat{m}_0$, and in Fig. 4-B2, the reconstructed conditioned probability. As expected, the constrained free energy, $\Omega$ here, presents 2 mininima at the two metastable states sampled before with SMC in Fig. 4-A, but their relative depth is significantly different for the SMC-RBM model. In the TMC-RBM case, the two minima have the same potential's value, meaning that they correspond to two states equally probable: the equilibrium distribution of the TMC-RBM match remarkably well with the dataset's distribution. Instead, in the case of the SMC-RBM, one minima is much higher than the other, and in fact, it is essentially suppressed in the equilibrium distribution $p(\hat{m}_0)$, see Fig. 4-B2. If we could wait long enough, all generated samples of the SMC-RBM would fall to the lower-$m_0$ minimum.

The conclusions of this simple experiment are rather worrying. First, when the clusters are well-separated in the learned RBM, the persistent chains of the classical training method remain trapped in one of the two clusters. The problem, here, is that it is impossible that the parallel chains are correctly balanced between the two minima of the potential. Still, since they can not escape from their local minimum, it gives the impression that the RBM is learning correctly the dataset while in reality the *true* equilibrium probability is not well-adjusted. This also means that the gradients cannot correct this problem. In fact, at longer training times, we observe that the SMC-RBM is not adjusting the location of the peaks correctly because of this effect.

We can illustrate this effect studying several intermediate models in the training of the SMC-RBM model, see Fig. 5-C where $t_{\text{age}}$ refers to the number of parameters updates performed. In this figure, we compare: (i) the equilibrium constrained measure obtained with the TMC method, and the histogram of the $m_0$ of the samples either in (ii) the persistent chain, or (iii) obtained with a long SMC sampling. We also include, next to each distribution, the trajectory followed by several independent Markov chains during the SMC sampling. We can see that at, and below $t_{\text{age}} = 91$, each Markov chain jumps several times from one cluster to another in 5000 MC sweeps, and because of that the TMC $p(\hat{m})$ matches very well the distribution of the persistent and generated samples, even when the distribution is bimodal. As the training progresses, for instance at $t_{\text{age}} = 96$ these jumps get rarer and rarer, which means

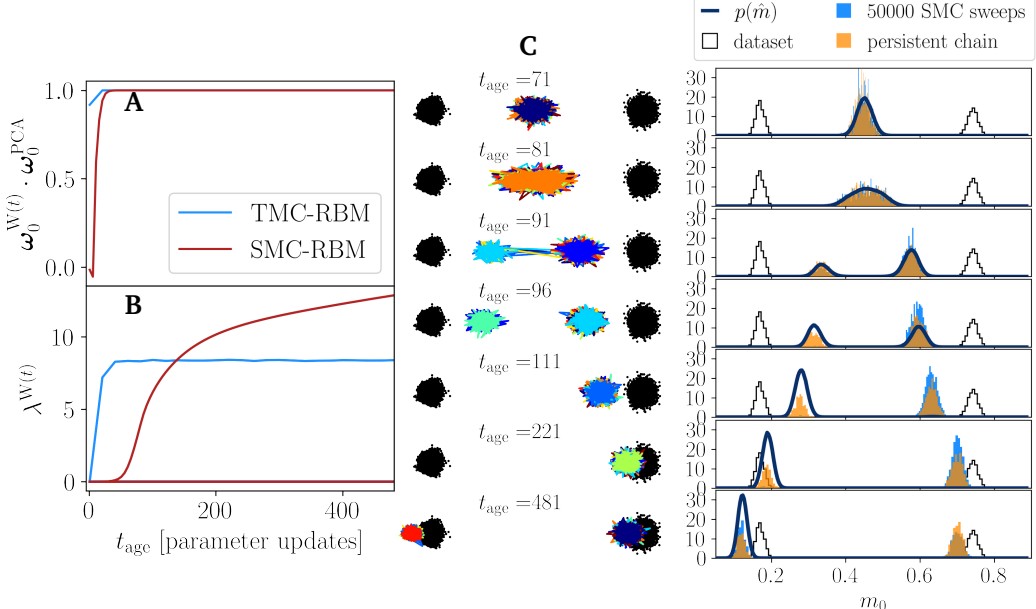

Figure 5: In **A** we show the projection of the first eigenvector of the RBM weight matrix $W$, $\boldsymbol{\omega}_0^{\mathrm{W}}$, onto the direction of the first principal component of the dataset $\boldsymbol{\omega}_0^{\mathrm{PCA}}$, as a function of the number of parameter updates $t_{\mathrm{age}}$, for a TMC-RBM and a SMC-RBM. In **B** we show the evolution of the first 10 eigenvalues of $W$ for the two machines. Only the first eigenvalue is expressed, since only one direction is needed for this dataset: the others are almost zero. In **C** we examine the SMC-RBM at 7 different stages of training ($t_{\mathrm{age}} = 71, 81, 91, 96, 111, 221, 481$ updates). In particular, in the right panel of C we compare the equilibrium distribution of $m_0$ obtained with the TMC method $p(\hat{m})$, the histogram of the dataset, the histogram of the persistent chain during learning, and the histogram of the samples generated with 50000 SMC sweeps. To illustrate the trapping phenomena, we show in the left panel of C the trajectory of several Markov chains (in the $m_0 - m_1$ space) obtained during the SMC generation process (in color) and compare them with the scatter plot of the dataset projections (black dots).

that the persistent chains get trapped in each of the states with the proportion reached the last time the standard sampling was able of thermalising. Such proportion remains fixed for all the rest of the training, even if the equilibrium distribution of the model is actually unimodal. Neither the samples generated with standard MC moves are reliable.

Fig. 5-C let us illustrate another troublesome effect. Since the distribution of the persistent or generated chains differ strongly from that of the equilibrium, the negative term of the gradient is very badly computed and the learning trajectory is strongly affected by this. In fact, since the persistent chain is almost correctly located, the gradient is probably only adjusting little details of the parameters, while in reality, the equilibrium samples of the true model do not correspond to these samples and therefore the RBM end up very badly trained. Indeed the normal evolution of the training moves towards increasing the relative probability of the low $m_0$ state (the persistent chain contained an over-representation of high $m_0$ samples), which only contributes to suppress more and more the probability of the second state.

A sign of this training problem can be easily detected in this simple dataset by monitoring the eigenvalues of the weight matrix during the learning, $\lambda_\alpha$. In this simple dataset, we observe that only the first eigenvalue $\lambda_0$ is turned on during the training. This $\lambda_0$ is associated to an

eigenvector, $\boldsymbol{\omega}_0^{\mathrm{W}}$, that in both kinds of machines, the TMC-RBM and the SMC-RBM, aligns perfectly with the first principal component of the dataset, $\boldsymbol{\omega}_0^{\mathrm{PCA}}$ in Fig. 5-A. Yet, the value of $\lambda_0$ does differ from one training to another. Indeed, for the TMC-RBM, the learning stops at a given moment and the first eigenvalue remains constant from then and on. While in the SMC-RBM scheme, the $\lambda_0$ keeps growing as the gradient is not computed correctly. This is translated in a decrease of the effective temperature of the model (and a sharpening of the equilibrium peak that is appreciated in the compact trajectories of the $t_{\mathrm{age}} = 481$ RBM in Fig. 5-C).

### 6.1.2 Two-dimensional dataset

We demonstrate again the feasibility of our approach for a case where the clusters span a two-dimensional subspace (see orange dots in Fig. 6). As in the 1D case, a SMC sampling fails to jump from one cluster to another one.

In this case, we rely on a 2D version of the TMC method. We need to constrain the magnetization along two principal directions because a single constraint is not able to split up the 3 clusters because they superimpose when projected on a single direction. A TMC-1D strategy would then suffer from convergence problems too (as the SMC). In this case, we show directly the results of the TMC approach, since it is clear from the previous example that the usual training algorithm cannot work. We show on Fig. 6 the TMC effective potential $\Omega$ extracted at various steps of the learning together with the projection of the dataset along the two chosen directions. On the last panel of Fig.6 (bottom-right) we plot the learning curves of the eigenvalues of the weight matrix. We see that, once the first two eigenvalues are learned, the machine stops evolving on linear timescale. Still, some more eigenvalues are increasing at large number of epoch indicating that we should probably lower the learning rate to stabilize the learning. We also show the evolution of the effective potential at different instants during the training process. We can appreciate how the potential adjust to fit the 3 different minima, and also how the final TMC reconstructed probability matches perfectly the distribution of the dataset in the $t_{\mathrm{age}} = 1001$ figure. This shows how this strategy works perfectly on these difficult datasets.

## 6.2 Real datasets

We now apply our TMC training method on real datasets. It is clear that big improvements are only expected in datasets that are clustered in some way. We therefore focus on two cases. First on the MNIST dataset reduced to the digits 0 and 1. It has been observed recently, that SMC samplings struggle to jump from one digit to another, making convergent generation samplings extremely long [34]. The second example is taken from the 1000 Human Genome Project [35], already studied in a generative context recently [14, 20]. In this last dataset, a strong clustered structure is observed when the data are projected along the first principal components of the dataset, $\{\boldsymbol{\omega}_\alpha\}$.

### 6.2.1 MNIST 0-1 dataset

This dataset was created by taking only the 0 and 1 digits from the MNIST dataset [36]. We obtain slightly more than $10^4$ images. Unlike the full dataset, this reduced dataset splits into two well-separated clusters when projected along $\boldsymbol{\omega}_0$ (the first principal direction of the dataset centered on zero), see Fig. 7.

In this example, we will again compare the performance of a TMC-RBM with that of a SMC-RBM. A critical moment in training this dataset is when the first principal direction is learned. In this regime, the phase space of the RBM consists of two separate modes where

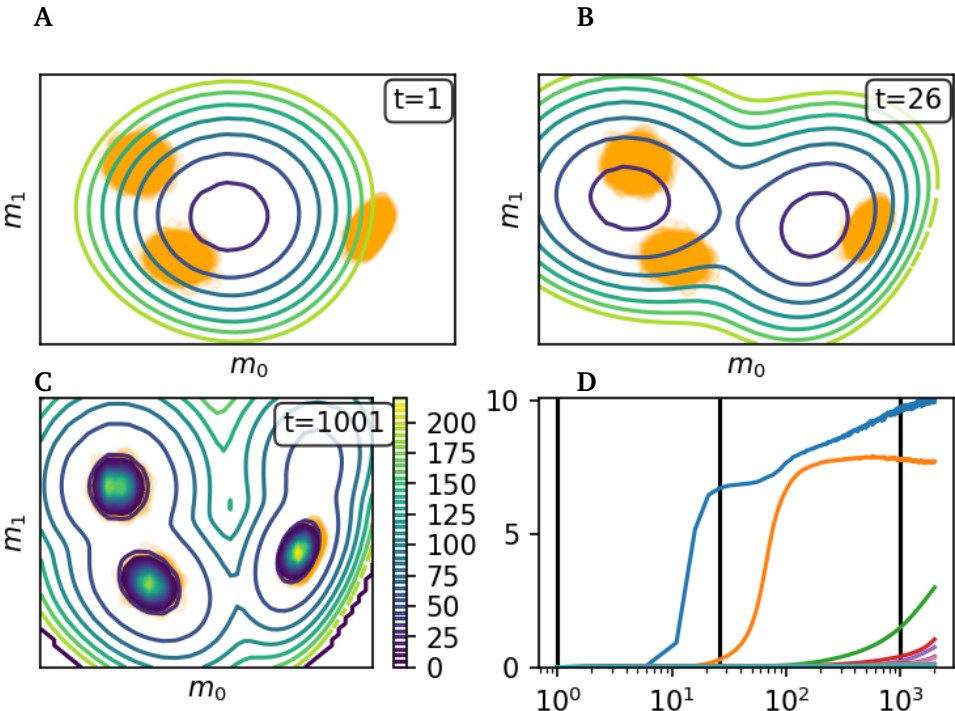

Figure 6: **A-C:** In orange dots, we show the $m_0$ and $m_1$ projections of the samples of a synthetic dataset of 3 clusters. In a contour map, we show the TMC effective potential $\Omega$ reconstructed at different moments of the training of a TMC-RBM ($t_{age}$ refers to the number of updates). At $t_{age} = 1001$, we also show the TMC probability distribution (in a heatmap), which correctly reflects the density of the three clusters. In **D** we show the evolution of the eigenvalues of the $W$ matrix as a function of $t_{age}$.

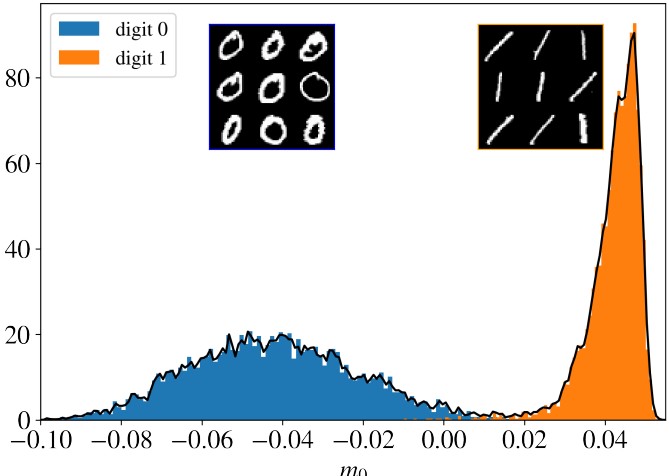

Figure 7: Histogram of the projection $m_0$ of all samples of the dataset MNIST 0-1 along its first principal component. We show in blue and orange the histogram of the magnetizations of only the 0 and 1 digits. It is clear that $m_0$ efficiently separates both digits.

an SMC dynamics has trouble jumping from one to the other, while TMC should have no problem. Later in training, as more and more directions are learned, we will see that the strong acceleration of TMC is damped because the trained RBM seems to create alternative dynamical paths to jump from 0s to 1s that do not have to overcome this large barrier in $m_0$. We can characterize these dynamical effects by computing the integrated autocorrelation times [37, 38], $\tau_{\text{int}}$, of the evolution of $m_0(t)$ in the two different sampling dynamics:

$$\tau_{\text{int}} = \frac{1}{2} + \sum_{t=1}^{\kappa \tau_{\text{int}}} \frac{\rho_m(t)}{\rho_m(0)}, \quad \text{with}$$

$$\rho_m(t) = \frac{1}{M-t} \sum_s^{M-t} [m_0(s) - \overline{m}_0][m_0(s+t) - \overline{m}_0],$$

where $\kappa$ is a small number, in our case $\kappa = 6$ as in [37, 38], and $\overline{m}_0$ is the empirical estimator of the equilibrium $\langle m_0 \rangle_{\mathcal{H}}$ using $M$ independent samples.

In practice, we find that the relaxation time of the SMC dynamics is enormous (not even measurable in "reasonable" times and higher than $10^7$) when only one direction was learned, while the TMC relaxes in only a few sweeps. We estimate the SMC relaxation time in each RBM in Fig. 8–B. This divergence of times is related to the apparition of two very distant metastable states, see Fig. 8–A, just as we discussed in the context of the artificial 1D dataset. Then, in a second learning phase, the SMC relaxation time falls back to measurable values using the transverse directions learned by the RBMs. Even though SMC thermalization times remain quite high, SMC sampling becomes competitive with TMC sampling in terms of physical computation times because TMC relaxation times grow rapidly when $m_0$ is no longer able to break metastabilities, and because TMC sampling is much slower than SMC because the update of visible variables cannot be parallelized. Nevertheless, quite unexpectedly, we find that TMC-RBMs have much faster SMC relaxational dynamics than SMC-RBMs and that the factor between the two increases with the number of parameter updates, see Fig. 8–B.

When we compare the samples generated with the two learned RBMs at $t_{\text{age}} = 1001$ parameter updates, we find that those obtained with the TMC-RBM are qualitatively better than those from the SMC-RBM. However, we emphasize that this comparison is difficult to see from a direct visual inspection, see Fig. 9A, since the difference is not related to the overall quality or definition of the digits, but rather to the ratio of 1s to 0s and the variability of the shape of the generated digits. The comparison becomes clearer if we project the generated data onto the first 4 principal components and compare the histograms obtained with those obtained with the MNIST projections, see Fig. 9C. It is clear that the equilibrium distribution of TMC-RBM reproduces much better the original. Another interesting effect is that the typical time needed to jump from one digit to another (from one cluster to another) during the generation sampling is faster for the TMC-RBM, as illustrated in Fig. 9B.

The reason why TMC-RBMs present faster relaxations than SMC-RBMs is not clear, but we think it is related to the effects of the nonequilibrium regime that were encoded in the model during training when the mixing times of the model were decades above the number of Gibbs steps used to train the machine, as estimated in Fig. 8B. Indeed, in Ref. [20] we observed that one of the consequences of RBMs trained deeply in non-equilibrium was precisely their extremely slow relaxation. This effect is also probably related to the degradation phase observed in Ref. [39].

### 6.2.2 Human genome dataset

We considered the population genetics' dataset of Ref. [35, 40]. This dataset corresponds to a sub-part of the genome of a population of 2504 individuals (each providing two strands of

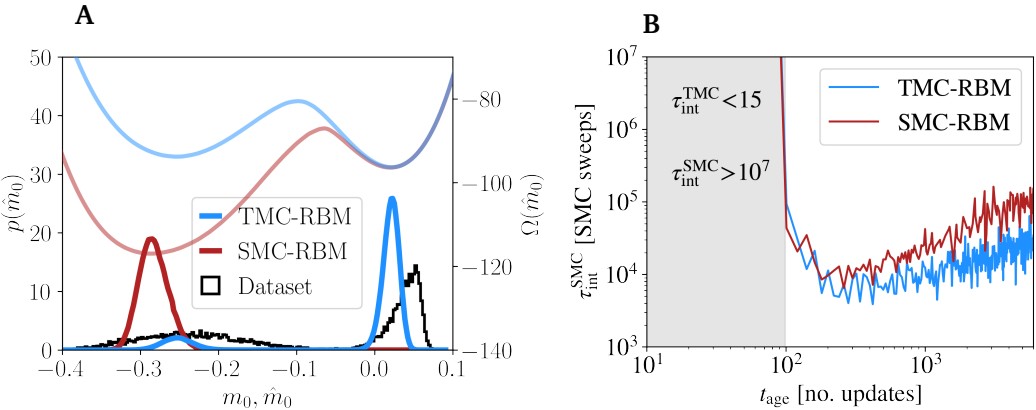

Figure 8: **A:** Histogram of the dataset along the first direction $\boldsymbol{\omega}_0$, together with the tethered potential $\Omega(\hat{m}_0)$ and the corresponding reconstructed TMC probability $p(\hat{m}_0)$ of both RBMs in the early stages of learning (here $t_{\text{age}} = 61$). We clearly observe the formation of two metastable states in both machines. In practice, we observe that Markov chains using SMC dynamics are trapped in one of these metastable states since initialization and cannot visit the rest of the phase space. Exactly as described in the synthetic 1D dataset. **B:** We estimate the SMC integrated autocorrelation time $\tau_{\text{int}}^{\text{SMC}}$ of $m_0(t)$ as a function of the number of training updates used to train the two machines: TMC-RBM and SMC-RBM. The shaded area marks the region where $\tau_{\text{int}}^{\text{SMC}}$ exceeds $10^7$ MC sweeps, while $\tau_{\text{int}}^{\text{TMC}}$ (for the slower $\hat{m}$ constraints ) remains below 15 MC sweeps. When $t_{\text{age}} \gtrsim 10^2$, the SMC relaxation time drops to feasible values in both machines. Nevertheless, the relaxation dynamics of TMC-RBM is significantly faster than that of SMC-RBM.

DNA, yielding a dataset of 5008 samples) sampled from 26 populations in Africa, East Asia, South Asia, Europe, and the Americas. Each variable is binary $\{0, 1\}$ and indicates whether or not a particular gene is altered compared to the human reference genome. A total of 805 alterations are reported, reflecting a high proportion of the population structure present in the whole genome dataset. The dataset is therefore made of 805 binary variables and 5008 independent samples. For our study, the dataset was made of 4500 samples for the training set. This dataset has a very clustered nature, as can be appreciated in the dataset's projection along the second and third principal directions shown in Fig. 10 where each dataset sample is represented as a black dot. One can also see that SMC dynamics get completely trapped in these different clusters in a well-trained machine, as we illustrate with different color trajectories each corresponding to an independent sampling of the RBM.

We therefore constrain these directions for the training of the RBM using the TMC method. For this case, we will exhibit two clear advantages of this method. First, we show the feasibility of training the RBM on a real dataset using a set of two constrains. In this setting, the TMC method is not only useful to estimate better the negative term of the gradient, but it also gives the possibility to monitor how-well the equilibrium measure of the RBM is matching the data through the TMC effective potential. It is worth of mentioning that evaluating the quality of an RBM (that is, ensuring a proper adjustment of the density peaks with those of the dataset) with traditional SMC sampling methods is extremely hard, because jumps between the different metastable clusters are very rare and there are four of such clusters involved.

In fact, on Fig. 11 we show the TMC effective potential and the reconstructed distribution at various stages of the learning. It is interesting to see that it seems quite difficult to adjust correctly the 4 clusters to their correct weight. Thanks to the TMC method, we can easily track the position of the RBM's clusters and diminish the learning rate to fine-tune the learning. On Fig. 11, we can see that after $t_{\text{age}} \approx 1200$, the TMC-RBM seems to be well-learned.

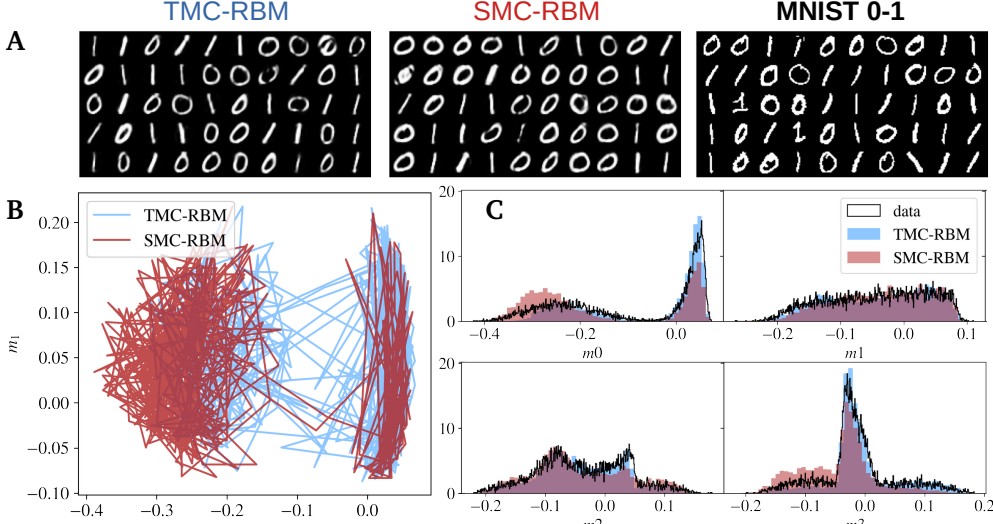

Figure 9: For a TMC-RBM and a SMC-RBM, both trained for 1001 parameter updates with the same learning rate $\gamma = 0.01$ and the same $k = 10$, we show: **A:** 50 equilibrium samples generated (with each model) and compared to 50 samples of the original MNIST dataset. In order to ensure equilibration, we performed $10^7$ SMC sweeps prior to imaging **B:** For both machines, we show the scatter plot of the SMC dynamical trajectories projected along the first two directions: $m_1(t)$, $m_0(t)$, all $10^3$ sweeps, for $5 \cdot 10^5$ SMC sweeps. It can be clearly seen that the sampling of the SMC-RBM is slower, since there are fewer jumps (red curve) between digits 0 and 1. **C:** The histogram of the data generated by the two RBMs (after $10^7$ SMC steps), in red by the SMC-RBM and in blue by the TMC-RBM, projected along the first four principal directions of the dataset. The black line corresponds to the histogram of the dataset. It is clear that the SMC-RBM has many more anomalies than the TMC.

The second crucial improvement is that, only with the TMC method we can guarantee that all the clusters are visited regularly. Indeed, we already showed in Fig. 10 that we suffered strong break of ergocity issues even after $10^5$ SMC sweeps. This fact tells us that, on this dataset, each persistent chain of the PCD will remain trapped in certain regions of the phase space and will not be representative of the equilibrium measure of the RBM. As a consequence, the persistent chains will not estimate correctly the negative term of the gradient.

Finally, it is possible to try to train this dataset using the classical PCD - SMC algorithm. However, what we see is that the learning dynamics has a lot of trouble adjusting correctly the different minima, and that in addition, even if one can "cherry pick" the exact number of updates where the trained SMC-RBM has all the good dataset minima, such a process requires using the TMC method to track the presence of the good minima in the effective potential because SMC samplings are completely stuck. Even in that case, we observe that again TMC-RBMs relax faster than the equivalent SMC-RBM (for the same $t_{age}$). Yet, SMC dynamics are that slow for both kinds of machines, that computing $\tau_{int}^{SMC}$, as done in Fig. 8 for MNIST 0-1s, is beyond our numerical capacities. Still we can compare for both machines, the relaxation time in the TMC runs (which are much shorter). We obtain that $\tau_{int}^{TMC}[SMC-RBM] \sim 4 \cdot \tau_{int}^{TMC}[TMC-RBM]$, showing the TMC-RBM is again faster that the PCD one.

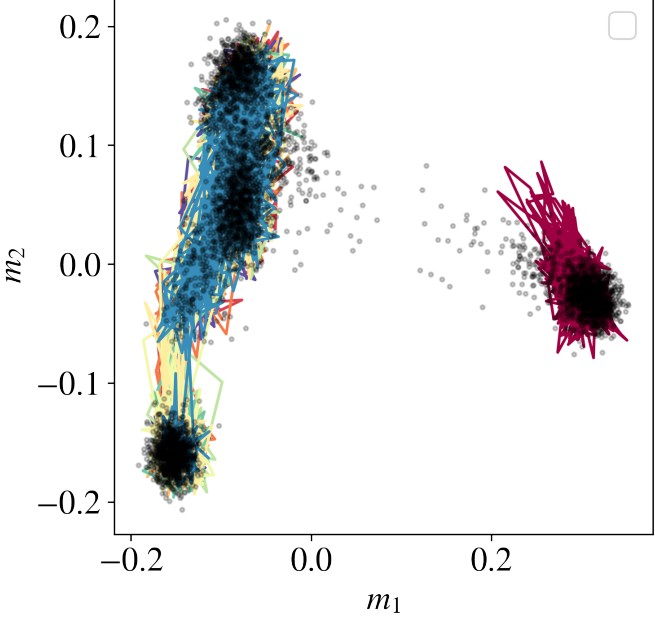

Figure 10: We show the training dataset projected along the second and third principal directions (namely $m_1$ and $m_2$) in black dots. The samples are grouped in separate clusters. In the colored solid lines, we show different SMC sampling trajectories $\{m_1(t), m_2(t)\}_t$ of $10^5$ MC sweeps extracted from a TMC-RBM trained for $t_{\mathrm{age}} = 1000$ updates, $\lambda = 0.01$ and $k = 10$. One can clearly appreciate that there is no jump between the high and low $m_1$ sectors during this period.

## 7 Runtime comparison

The TMC method has two major disadvantages. First, the global constraint imposed on the measure breaks the conditional independence of the two layers: If the constraint applies to the visible nodes, the hidden nodes are still independent given a set of visible nodes, but the converse is no longer true. Because of the constraint, the visible nodes are no longer independent once the hidden nodes are fixed, since there is an explicit interaction between the visible nodes. Therefore, we cannot create a new state for the visible nodes in parallel, which means that part of the advantage (speed increase) of using GPUs is lost. Another disadvantage of the TMC method is that we no longer have to calculate the correlations between the visible and hidden nodes for a single RBM, but for as many discretization step RBMs as we used to constrain the order parameters. However, once the machine is learned, the time required to generate samples with TMC is significantly reduced compared to SMC. For each generated data point, the process is as follows: first, a TMC constraint $\hat{m}$ is randomly drawn from the reconstructed probability $p(\hat{m})$, and then a TMC sampling process is performed at fixed $\hat{m}$, a dynamical MC process that converges to equilibrium in a very short time thanks to the constrained ensemble. In Fig. 12 we show how the runtimes compare when training the RBM with TMC or SMC and on CPU or GPU. The SMC-GPU is much faster than all other implementations, making it the perfect candidate when the phase space if not clustered. Still, when an energy barrier is present, the only mean to reach thermalization in a large system is to use the TMC approach. As future improvement, it shall be possible to tether the hidden magnetization (rather than the visible one). In that case, the performance bottleneck would be reduced since the number of hidden nodes is generally much smaller than the number of visible ones, in particular in low-dimensional datasets.

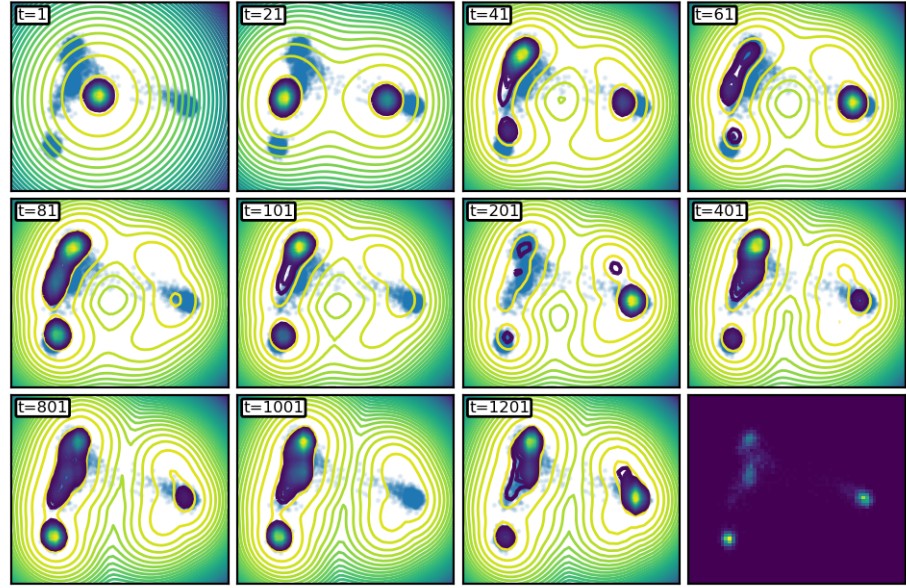

Figure 11: Evolution of the potential along the second and third principal directions during the learning of the TMC-RBM. We clearly observe how the second direction is learned by the RBM, followed by the third. After one thousand parameter updates, the reconstructed probability distribution covers the dataset.

## 8 Conclusion

We have demonstrated the feasibility and benefits of using a biased Monte Carlo sampling strategy for both training — i.e., to better estimate the negative terms of the gradient in order to update the parameters of the model — and for generating new samples in trained RBMs. The largest improvements are observed for highly clustered datasets, where classical local MCMC moves get stuck in one or several of these clusters and fail to sample ergodically the entire phase space (in reasonable time). We show that the Tethered Monte Carlo method is extremely efficient for sampling the equilibrium measure of these types of models, as long as one can find good observables to uniquely identify each cluster. We show that the projection along the first PCA directions in the early stages of training are good order parameters as motivated from the mean-field results [16,32]. We show that this TMC strategy is crucial to correctly learn artificial low-dimensional clustered datasets. On these datasets, the standard learning approach that combines parallel local MCMC updates with PCD completely fails to obtain reliable models for the data. On real datasets, we find that as learning progresses, more ordering parameters are needed to avoid the presence of metastabilities during sampling, which means that the TMC strategy with only one or two constraints also starts to suffer from ergodicity problems. We note, however, that even if TMC does not thermalize easily, the models trained with TMC exhibit significantly faster relaxation dynamics than the models obtained with the standard approach.

Moreover, we find that even for RBMs trained with standard sampling techniques, the TMC sampling approach is extremely useful to assess the quality of a given trained machine, since it allows a direct calculation of the probability distribution of the model, projected along a set

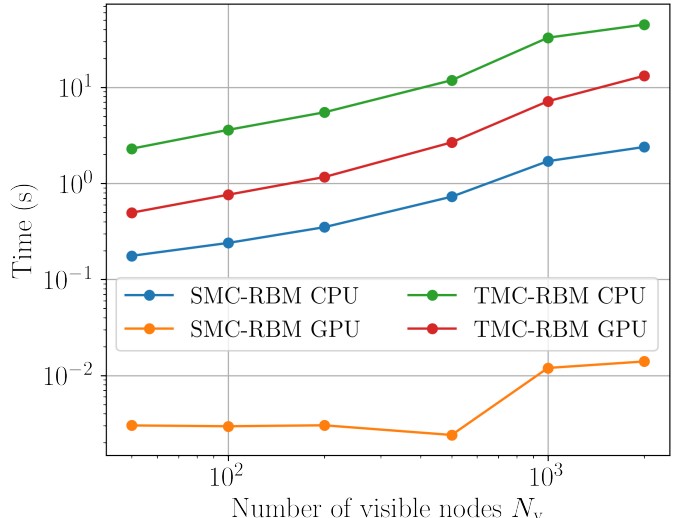

Figure 12: We show the average wall time per epoch (averaged over 20 epochs) for RBMs trained on either CPU/GPU and using the TMC or SMC method during learning. The curves were generated with a Geforce RTX 3090 as GPU and an AMD Ryzen 9 5950X as CPU.

of selected directions, thus providing the possibility to estimate how well the distribution of the model matches the empirical one. The developed approach is thus a practical alternative to the most commonly used Gibbs sampling algorithm when the intrinsic properties of the dataset drive learning toward a model with well-separated modes.

Despite the impressive performance of the TMC model on clustered datasets, it has two important drawbacks: First, it is much slower than the traditional alternative sampling method in terms of computation time because the global constraint introduces correlations between the visible (or hidden) variables, thus removing the conditional independence of the two layers that favors parallelization. For this reason, alternative approaches involving only local constraints should be considered in the future to restore parallel performance. A possible improvement could be to constrain the value of some hidden nodes with a condition that does not remove the conditional independence. However, a clever mechanism must be found to select the hidden nodes to be fixed. Second, the dimensionality of the constraints. Since we need to integrate the potential exactly, we need to discretize the space into as many dimensions as the number of constraints allows. This makes the method cumbersome even with three constraints and probably impossible with more.

# Acknowledgments

This work was supported by the Comunidad de Madrid and the Complutense University of Madrid (Spain) through the Atracción de Talento programs (Refs. 2019-T1/TIC-13298 for A.D. and 2019-T1/TIC-12776 for B.S.), the Banco Santander and the UCM (grant PR44/21-29937); and Ministerio de Economía y Competitividad, Agencia Estatal de Investigación and Fondo Europeo de Desarrollo Regional (FEDER) (Spain and European Union) through the grants. PID2021-125506NA-I00 and PGC2018-094684-B-C21. N.B. would like to thank the Department of Theoretical Physics of the Complutense University of Madrid for the kind reception and support during his stay.

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
