# Peer review of "Learning a Restricted Boltzmann Machine using biased Monte Carlo sampling"

_SciPost Physics, doi:SciPost Phys. 14, 032 (2023)_

## Round 1 · Referee Report · Anonymous (Referee 1) · 2022-7-5

Strengths

1- Novel approach for sampling from and training Restricted Boltzmann Machines based on the Tethered Monte Carlo biased sampling algorithm. 2- Tackles an important limitation of RBMs: its inability to reliably learn from clustered datasets 3- Well-grounded and innovative approach 4- Thoroughly benchmarked on artificial and real datasets 5- Limitations are clearly stated and investigated in the main text

Weaknesses

1-Proposed approach is slow, as there is no efficient way to sample from the Tethered distribution $P(v,h | \hat{m})$. 2- No runtime comparison between proposed TMC sampling algorithm and current state-of-the-art sampler (parallel Gibbs). 2- Impossible to tract more than 2-3 order parameters. 3- MCMC equilibrium time may still diverge due to energy barriers in orthogonal directions.

Report

Restricted Boltzmann Machines (RBM) are simple two-layer neural networks that jointly learn a probability distribution and a representation of data. RBMs can faithfully model complex probability distribution despite their simple architecture and are highly interpretable. However, training RBMs requires evaluating the moments of the distribution, which is usually done by MCMC sampling. MCMC sampling is especially challenging for multimodal data sets, as in this case Markov Chain have very slow equilibration time.

Here, Béreux et al. propose to use a biased Monte Carlo sampling scheme originally introduced for physical systems with multiple coexisting phases. The Tethered Monte Carlo method (TMC) consists in:
1. Introducing one or few auxiliary variables $\hat{m}$ coupled to the original system, such that $< \hat{m} | v,h > = m(v)$ with $m(v)$ one order parameter computed from data (here, a principal component) and $Var( \hat{m} | v,h) = 1/\alpha$ and $\alpha$ is a (large) tunable parameter.
2. Repeatedly sampling from the conditional distribution $P(v,h | \hat{m} ) $ for values of $\hat{m}$ covering all possible values.
3. Determining the marginal $P( \hat{m})$ by numerical integration of
4. Estimating moments of the original distribution using the law of total probability

The rationale being that often, the conditional distribution is easier to sample from than the original distribution. The authors test their approach in sampling-only and full learning settings on i) 1D and 2D artificial datasets ii) the MNIST-0/1 and iii) human genome dataset.

They find that for simple multimodal data, samples generated from the TMC sampler faithfully represents the true, multimodal distribution (unlike naive Monte Carlo), leading to a successful training. For realistic data, the TMC sampler significantly improves upon the naive sampler, but the authors note that equilibrium times can still diverge due to slow mixing along directions orthogonal to the tethered ones. The practical limitations of the approach (as stated by the authors) are: i) Lack of efficient, factorized sampling for the conditional distribution and ii) Inability to tether more than 2-3 order parameters.

The proposed approach is innovative, well-implemented and carefully benchmarked. In principle, it efficiently solves the obnoxious problem of sampling for “vanilla” clustered datasets. Limitations of the approach for real-life datasets are properly investigated in the main text (but not in the abstract, see below). My main concern is that, as stated by the authors, the approach is too slow for large-scale data where the number of variables is ~ 1k-10k. CPU run-time comparison between SMC and TMC should be added in the paper for clarity. Also, it is important that the speed/applicability limitations are stated in the abstract. The following claim in the abstract is not sufficiently substantiated:

"This sampling technique solves efficiently the problem of evaluating the quality of a given trained model and the generation of new samples in reasonable times. "

Altogether, I recommend publication with minor revisions.

One suggestion: Did the authors experiment on tethering one or few hidden units rather than the principal components ? Sampling from the conditional P(v,h | \hat{m} ) would be substantially easier as conditional independence would be preserved. Moreover, hidden units may play a similar role as principal components, as they learn the main collective modes of variation of data and thus have slow dynamics. It should be possible to dynamically pick the tethered hidden units using heuristics such as weight norm / hidden unit importance or based on the autocorrelation function of their marginal. If two or three hidden units are chosen, they should be uncorrelated to maximize diversity.

Requested changes

1- Please provide runtime comparisons for TMC against SMC. 2- Please state limitations of the proposed approach in the abstract 3- Figure 6D: it seems that the training has not converged after 2K updates. Please show more training points to validate convergence or discuss potential issues. 4- Please define in the paper the integrated autocorrelation time tau_int rather than referring the reader to a 500+ pages text book. 5- Typo p16: “just as we discussed in the contest of the artificial 1D dataset.” 6- Labels of Figure 9A seem swapped. Please check. 7- Spelling p19: “ minima, and that in addition, even if one can “cherry peak" the exact number of updates where”

---

## Round 1 · Referee Report · Anonymous (Referee 2) · 2022-7-10

Strengths

(1) The paper recalls well the context and exemplifies the corner stone issue with classical methods in simple, pedagogical examples.
(2) The experiments are presented with significant detail allowing to better understand the inner workings and advantages of the algorithm.
(3) The proposed method enables both the learning of RBMs in multimodal cases (a.k.a. clustered datasets) but also offers new opportunities in evaluating the quality of training of these generative models.

Weaknesses

(1) The proposed method can only apply when one can identify a few dimensions that will separate the modes.
Minors:
(2) Some minor details are not entirely clear in the numerical sections (see report below) and the text of this result section could a bit synthesized to ease the reading.

Report

The paper proposes a detailed demonstration of the application of a biased MCMC for the training of RBMs. Namely, the authors suggest using the principal components of the training set to guide a tethered Monte Carlo when computing the gradients of the log-likelihood. The paper is clearly written and presents convincing experiments to demonstrate that the method can be applied in practice - within its domain of application. Indeed, as pointed out by the authors, the limitation of the proposed method lies in (1) the type of structure required in the data and (2) the ability to identify this structure from the training set. Although one can expect that these requirements are stringent and will not be met in most use-cases, I would argue that the present paper raises an important question and offers a relevant discussion: few works have discussed the equilibration issue of MCMC in RBM training.

Requested changes

Questions for the authors: - It is mentioned that MNIST images are qualitatively better - could the author back this claim by including such a figure? - Do they authors understand why the TMC trained RBMs seem to have shorter relaxation times for SMC algorithms? - In particular for MNIST, do the author understand why the TMC trained RBMs is well sampled by an SMC? Does this suggest the MNIST 0-1 modes are actually not that separated? Would that also be the reason why it is usually considered that MNIST can be learned with CD or PCD by an RBM (as the authors demonstrate that the multi-modality a priori prohibits any learning with traditional MCMCs for clustered datasets)?

Minor: - There seems to be a notation change at the bottom of page (8) where the hidden variables are denoted by \tau. - Also, what does “permanent chain” refer to? Is it the “persistent chain”of PCD? - Maybe a terminology slightly confusing is the employed “low-dimensional datasets”, “6.1.1. one-dimensional dataset”. Actually it is not entirely clear whether the considered datasets have actually low-dimensional subspace support embedded in high dimension, or whether the only assumption is that a projection onto a low-dimensional subspace is enough to differentiate the clusters. Can the authors clarify? - In figure 5B, why are there two red lines? - In figure 5C, what does the black points represent? The dataset? - Taking only the 0-1 in MNIST should yield about 10,000 images rather than 10ˆ5 (page 16). - Unless I missed something, it would be useful to describe more the human genome dataset somewhere in the paper. Are the data binary? In which dimension? Is there one variable per gene?

---

## Round 2 · Referee Report · Anonymous · 2022-10-6

Report

All my points have been thoroughly addressed. I support publication as is.

---

## Round 2 · Referee Report · Anonymous · 2022-10-17

Report

I think the authors for their careful answers to my questions and for taking into account my comments in their revision.

I recommend acceptance in the current form.

---

## Round 2 · Author Response

Dear Editor,

Please find below the revision of our manuscript "Learning a Restricted Boltzmann Machine using biased Monte Carlo sampling", which we hereby submit for a second review. We would like to thank the reviewers for their careful and constructive reading of the previous version of the manuscript, and for their positive feedback and numerous comments and suggestions. In the following, we address all their comments and suggestions point by point.

Comments of the first referee

  1. It is mentioned that MNIST images are qualitatively better - could the author back this claim by including such a figure?

Answer: This claim is supported by Figure 9B, which compares the histograms of the projections of the data generated by the two machines with the histogram of the original data set. The histograms of the samples generated by the TMCRBMs are more similar to the data set than those generated by the SMCRBMs. As for the actual images, when examined visually, it is hard to say. Both machines produce good looking digits, only the SMC RBM tend to produce a greater imbalance between 0s and 1s and less variability. We have included several images of the equilibrium configurations obtained with each machine in Fig.9A.

  1. Do they authors understand why the TMC trained RBMs seem to have shorter relaxation times for SMC algorithms?

Answer: We have at least an empirical explanation. In a previous work [1], we observed that RBMs trained with nonconvergent MC chains to estimate the negative term of the gradient operate in the so-called nonequilibrium regime. These machines learn to encode the dynamical process they were trained with, rather than reproducing the statistics of the data set in their equilibrium measure. These machines with memory tend to have extremely slow dynamics because they are optimized to produce "good samples\" at some stage before reaching equilibrium. We further observed that RBMs trained with Rdm and some few steps ($k \sim 10$), produced RBMs with extremely slow dynamics, such as never ending aging effects. However, RBMs trained with PCD are more difficult to analyze because, first, the chain initialization statistics for the generation samples cannot be easily retrieved, and second, the effective number of Gibbs steps increases during training as the parameters change more slowly as training progresses. PCD RBM tends to favor equilibrium regimes but eventually falls out of equilibrium if the mixing times diverge, as is the case with the very clustered datasets. This means that out-of-equilibrium phenomenology is expected at times greater than the effective number of sampling steps of the persistent chain during training, which can be very long if the RBM has been trained for a very long time. Recent work has shown that upon training, RBMs eventually enter a degradation regime in which the mixing times of the model begin to diverge [2]. This degradation regime appears to be analogous to what we observed when we forced non-equilibrium sampling for too long during training. This degradation regime is then expected to arrive later in TMCRBMs because one can guarantee equilibrium for longer epochs. We have included this discussion in the new version of the manuscript.

  1. In particular for MNIST, do the author understand why the TMC trained RBMs is well sampled by an SMC? Does this suggest the MNIST 0-1 modes are actually not that separated? Would that also be the reason why it is usually considered that MNIST can be learned with CD or PCD by an RBM (as the authors demonstrate that the multi-modality a priori prohibits any learning with traditional MCMCs for clustered datasets)?

Answer: Again, we rely mainly on intuition and empirical facts. Our intuition is that MNIST can be decently trained because, at least in the PCA space, all the images are somehow connected (as compared with the previous artificial datasets were clusters were completely separated already at the level of the PCA). The MNIST 0-1 is somewhat different, where only the first direction clearly separates the two types of digits. Nevertheless, this MNIST 0-1 data cannot be reduced to just one or two dimensions. In the first stage of learning, when the RBM has learned only the first direction, the classical SMC cannot correctly sample the model (as we show in Fig.8B). As more directions are learned, it is possible that the SMC will use other routes with lower barriers to switch between the different digits to explore the entire phase space.

Minor comments:

  1. There seems to be a notation change at the bottom of page (8) where the hidden variables are denoted by $\tau$.

Answer: Thank you for noticing it. Now it's corrected.

  1. Also, what does “permanent chain” refer to? Is it the “persistent chain”of PCD?

Answer: We agree with the referee, we changed permanent to persistent in the new version to avoid confusion.

  1. Maybe a terminology slightly confusing is the employed “low-dimensional datasets”, “6.1.1. one-dimensional dataset”. Actually it is not entirely clear whether the considered datasets have actually low-dimensional subspace support embedded in high dimension, or whether the only assumption is that a projection onto a low-dimensional subspace is enough to differentiate the clusters. Can the authors clarify?

Answer: The two artificial datasets are created along a line and a plane, respectively. In this sense, we mean that they are low-dimensional. However, due to their inherent noise, they exhibit fluctuations in the other dimension as well. We have added a sentence in the manuscript to make this clearer.

  1. In figure 5B, why are there two red lines?

Answer: There were other eigenvalues on the figure that do not grow during learning and therefore appear as a straight line. We have now pointed out in the caption that the first 10 eigenvalues are shown.

  1. In figure 5C, what does the black points represent? The dataset?

Answer: They correspond to the dataset projected in the $m_0-m_1$ space. We have now added a sentence in the caption of the figure clarifying this.

  1. Taking only the 0-1 in MNIST should yield about 10,000 images rather than 10ˆ5 (page 16).

Answer: Indeed, that was a mistake on our part. We have corrected the amount in the new version.

  1. Unless I missed something, it would be useful to describe more the human genome dataset somewhere in the paper. Are the data binary? In which dimension? Is there one variable per gene?

Answer: We agree with the referee that the description of the dataset was too brief. A new paragraph about this dataset has been amended at the beginning of the section to be more precise.

Comments of the second referee

  1. Please provide runtime comparisons for TMC against SMC.

Answer: We have added a new section "Runtime Comparison" to discuss the runtime of different methods on different architectures.

  1. Please state limitations of the proposed approach in the abstract

Answer: We have added a sentence discussing the limitations in the new abstract.

  1. Figure 6D: it seems that the training has not converged after 2K updates. Please show more training points to validate convergence or discuss potential issues.

Answer: The full-convergence of RBMs is a complicated matter and eigenvalues do not generally converge. Empirically one finds that eigenvalues grow (slowly) forever unless certain regularization is introduced (or just one eigenvalue is necessary as in Fig.5A). In fact, it has been shown in [3] that the gradient with respect to the rotation of the eigenvectors $u^\alpha$ and $v^\alpha$ are quite unstable when two eigenmodes $w^\alpha$ and $w^\beta$ are getting closer. In the experiment corresponding to fig. 6D, we can first observe that the evolution is quite slow (the x-axis is in log-scale) and the slow-increasing trend that we see is probability due to a mix of instability due to various reasons: the form of the gradient but also the noise due to the approximation when computing the negative term.

  1. Please define in the paper the integrated autocorrelation time $\tau_{\rm int}$ rather than referring the reader to a 500 pages text book.

Answer: We have added to the definition in the manuscript and included a new reference to lecture notes on Monte Carlo sampling.

  1. Typo p.16: “just as we discussed in the contest of the artificial 1D dataset.”

Answer: Corrected.

  1. Labels of Figure 9A seem swapped. Please check.

Answer: Thank you for your comment, this error has gone unnoticed. The colors and caption in the figure are now corrected.

  1. Spelling p19: “ minima, and that in addition, even if one can “cherry peak" the exact number of updates where”

Answer: Corrected.

One suggestion:

Did the authors experiment on tethering one or few hidden units rather than the principal components ? Sampling from the conditional $P(v,h | \hat{m} )$ would be substantially easier as conditional independence would be preserved. Moreover, hidden units may play a similar role as principal components, as they learn the main collective modes of variation of data and thus have slow dynamics. It should be possible to dynamically pick the tethered hidden units using heuristics such as weight norm / hidden unit importance or based on the autocorrelation function of their marginal. If two or three hidden units are chosen, they should be uncorrelated to maximize diversity.

Answer: We thank the referee for the suggestion. It would be computationally convenient to tether just one hidden node, but it is not clear a priori which particular variable to choose. In addition, the RBM "suffers" from permutation symmetry of the hidden nodes, which makes it particularly difficult to make a guess about the index of the hidden node that might encode the relevant information without knowing in advance the relevant feature describing the data. Although the referee provides good suggestions about how to select the hidden nodes, we believe that this would be a significant change to the method, since it would mean adapting the code for such a task (which implies tuning the meta-parameters) and applying it to the paper's datasets. In any case, we add a sentence in the perspectives for future work.

References:

[1] Aurélien Decelle, Cyril Furtlehner, and Beatriz Seoane. Equilibrium and non-equilibrium regimes in the learning of restricted boltzmann machines. In A. Beygelzimer, Y. Dauphin, P. Liang, and J. Wortman Vaughan, editors, Advances in Neural Information Processing Systems, 2021.

[2] Lennart Dabelow and Masahito Ueda. Three learning stages and accuracy–efficiency tradeoff of restricted boltzmann machines. Nature communications, 13(1):1–11, 2022.

[3] Aurélien Decelle, Giancarlo Fissore, and Cyril Furtlehner. Thermodynamics of restricted boltzmann machines and related learning dynamics. Journal of Statistical Physics, 172(6):1576–1608, 2018

---

## Round 2 · List of Changes

### **List of changes:**

All major and minor comments of the referees have been taken into account.
In particular, we added
- a new section to compare the runtime of the different methods, including a new figure.
- a panel to fig. 9 to compare qualitatively the results.

Finally, the introduction and abstract have been polished.

---

## Editorial Decision

published